# Glycan-Dependent and -Independent Dual Recognition between DC-SIGN and Type II Serine Protease MSPL/TMPRSS13 in Colorectal Cancer Cells

**Motohiro Nonaka [1,2,†], Shogo Matsumoto [1,†], Bruce Yong Ma [1,3], Hiroshi Kido [4], Nana Kawasaki [5,6], Nobuko Kawasaki [1] and Toshisuke Kawasaki [1,*]**

[1]  Research Center for Glycobiotechnology, Ritsumeikan University, 1-1-1Nojihigashi, Kusatsu, Shiga 525-8577, Japan; nonaka.motohiro.4r@kyoto-u.ac.jp (M.N.); sho5pine@gmail.com (S.M.); czgsbym@yeah.net (B.Y.M.); 14v00048@gst.ritsumei.ac.jp (N.K.)

[2]  Department of Biological Chemistry, Human Health Sciences, Graduate School of Medicine, Kyoto University, 53 Shogoin Kawahara-cho, Sakyo-ku, Kyoto 606-8507, Japan

[3]  ZonHon Biopharma Institute, Inc., 518 Yunhe Road, Changzhou 213125, China

[4]  Division of Pathology and Metabolome Research for Host Defense, Institute for Enzyme Research, Tokushima University, 3-18-15 Kuramoto-cho, Tokushima 770-8503, Japan; kido@tokushima-u.ac.jp

[5]  Division of Biological Chemistry and Biologicals, National Institute of Health Sciences, Tokyo 158-8501, Japan; nana@yokohama-cu.ac.jp

[6]  Department of Medical Life Science, Graduate School of Medical Life Science, Yokohama City University, 1-7-29 Suehiro-cho, Tsurumi-ku, Yokohama 230-0045, Japan

*  Correspondence: tkawasak@fc.ritsumei.ac.jp; Tel.: +81-77-561-3444

†  These authors contributed equally to this work.

**Abstract:** A class of glycoproteins such as carcinoembryonic antigen (CEA)/CEA-related cell adhesion molecule 1(CEACAM1), CD26 (DPPIV), and mac-2 binding protein (Mac-2BP) harbor tumor-associated glycans in colorectal cancer. In this study, we identified type II transmembrane mosaic serine protease large-form (MSPL) and its splice variant transmembrane protease serine 13 (TMPRSS13) as ligands of Dendritic cell-specific intercellular adhesion molecule-3-grabbing nonintegrin (DC-SIGN) on the colorectal cancer cells. DC-SIGN is a C-type lectin expressed on dendritic cells, serves as a pattern recognition receptor for numerous pathogens such as human immunodeficiency virus (HIV) and M. tuberculosis. DC-SIGN recognizes these glycoproteins in a $Ca^{2+}$ dependent manner. Meanwhile, we found that MSPL proteolytically cleaves DC-SIGN in addition to the above glycan-mediated recognition. DC-SIGN was degraded more efficiently by MSPL when treated with ethylenediaminetetraacetic acid (EDTA), suggesting that glycan-dependent interaction of the two molecules partially blocked DC-SIGN degradation. Our findings uncovered a dual recognition system between DC-SIGN and MSPL/TMPRSS13, providing new insight into the mechanism underlying colorectal tumor microenvironment.

**Keywords:** C-type lectin; serine protease; colorectal cancer; tumor-associated glycan; DC-SIGN

## 1. Introduction

Dendritic cell-specific intercellular adhesion molecule-3-grabbing nonintegrin (DC-SIGN), a tetrameric transmembrane C-type lectin, which requires calcium for glycan recognition, is expressed in various types of human dendritic cells (DCs), including lymphoid DCs, peripheral blood monocyte-derived DCs (MoDCs), dermal DCs, and DCs in the mucosal epithelium [1–4]. In recent years, DC-SIGN has emerged as a key pattern recognition receptor involved in host defenses against

a wide variety of exogenous pathogens, including human immunodeficiency virus (HIV-1) [5], hepatitis C virus [6], Dengue virus [7], M. tuberculosis [8–10], and parasites [11]. Although DC-SIGN facilitates the CD8$^+$ and CD4$^+$ responses by promoting antigen uptake, processing, and presentation on MHC-I/MHC-II molecules, it also helps HIV and M. tuberculosis to subvert DC function and thus escape from intracellular degradation [8,12]. Indeed, studies of CD209 promoter gene polymorphism demonstrated that the -336 G variant allele, which is linked to lower DC-SIGN expression, is associated with decreased risk of Dengue fever [7] and tuberculosis [13]. DC-SIGN reportedly activates the serine/threonine kinase Raf-1 upon ligand stimulation and suppresses Toll-like receptor (TLR)-induced immune responses [14,15].

The interaction of DC-SIGN with pathogens is mediated by its carbohydrate recognition domain (CRD) in a Ca$^{2+}$-dependent manner. DC-SIGN shows specificity to mannose- and fucose-containing glycans and high affinity to high-mannose and fucose-containing Lewis (Le) glycans [5,16]. DC-SIGN recognizes pathogens that are heavily decorated with those glycans as non-self ligands. For example, lipoarabinomannan (ManLAM) is a major lipoglycan covering the cell wall of M. tuberculosis, with a structure that is not present in the host. DC-SIGN binds to ManLAM capped with dimeric and trimeric mannose residues but does not bind single mannose residues [10]. It has been proposed that the high binding affinity of DC-SIGN is achieved through tetramerization of DC-SIGN. Recognition of ManLAM by DC-SIGN inhibits DC maturation and induces strong upregulation of immune-inhibitory IL-10 production [15].

In addition to the recognition of foreign antigens, DC-SIGN is involved in the recognition of cancer cells through newly synthesized tumor-associated "non-self" glycans, as we and another group have demonstrated [16–18]. We found that DC-SIGN recognizes colon cancer cell lines (e.g., SW1116 and COLO205) and human colorectal cancer tissues based on Le (Le$^a$/Le$^b$) glycans [17,18]. Conditioned medium from MoDC-COLO205 co-cultured cells blocked MoDC maturation and attenuated TLR4-mediated immune activation. Considering that the ligand recognition step carried out by DC-SIGN regulates the subsequent induction of immunosuppressive responses, elucidating the mechanism of the DC-SIGN glycan recognition system is critical for the development of anti-cancer therapeutics. To date, carcinoembryonic antigen (CEA), CEA-related cell adhesion molecule 1 (CEACAM-1), and mac-2 binding protein (Mac-2BP) have been identified as endogenous DC-SIGN ligands carrying Le glycans [16–18].

In this study, we identified mosaic serine protease large-form (MSPL) and its alternative splicing variant transmembrane protease serine 13 (TMPRSS13) [19], which are type II transmembrane serine proteases [20–22], as novel DC-SIGN ligands. The binding of DC-SIGN to MSPL/TMPRSS13 was mediated by N-glycans of MSPL/TMPRSS13. Meanwhile, the soluble recombinant extracellular domain of DC-SIGN (DC-SIGN–ECD) was cleaved via MSPL/TMPRSS13 protease activity, indicating a different mode of recognition of DC-SIGN by MSPL/TMPRSS13. In the absence of Ca$^{2+}$, MSPL digested DC-SIGN more efficiently, suggesting that the molecular association mediated by N-glycans impeded DC-SIGN digestion. Clarification of the dual recognition processes between DC-SIGN and MSPL/TMPRSS13 may lead to the development of a treatment that efficiently suppresses colorectal cancer.

## 2. Materials and Methods

### 2.1. Cell Culture and Preparation of Recombinant Proteins

Human embryonic kidney HEK293 cells, human colon cancer COLO205 cells, and human monocyte U937 cells were obtained from the American Type Culture Collection. Human hepatoma HLF cells were obtained from the Japanese Collection of Research Bioresources cell bank. HEK293 and HLF cells were cultured in Dulbecco's modified Eagle's medium (Wako, Osaka, Japan), and COLO205 and U937 cells were cultured in RPMI-1640 (Wako) containing 10% fetal bovine serum at 37 °C with 5% CO$_2$. DC-SIGN-expressing U937 cells (U937-DC-SIGN) were generated through transfection with the pcDNA3-DC-SIGN plasmid [17] using Lipofectamine 2000 (Invitrogen, Carlsbad, CA, USA).

Stable transfected cells were selected with 1 mg/mL G418 (Invitrogen). Full-length MSPL and TMPRSS13 were each subcloned into the p3XFLAG-CMV plasmid previously [19]. The plasmids were transfected in HEK293 cells and their expression was confirmed through western blotting using anti-FLAG M2 antibody (Sigma-Aldrich, St. Louis, MO, USA). We subcloned cDNA for the extracellular domains of DC-SIGN (DC-SIGN–ECD) and MSPL (MSPL–ECD) into the p3XFLAG-CMV plasmid. The plasmids were transfected into HLF cells and selection was conducted using G418 at a concentration of 1 mg/mL. Stable transfected cells of DC-SIGN–ECD and MSPL–ECD were grown in ASF104 serum-free medium (Ajinomoto, Tokyo, Japan). The soluble recombinant proteins were purified with an affinity column of anti-FLAG M2 antibody.

### 2.2. DC-SIGN-Fc Affinity Chromatography

Purification of the membrane fraction of COLO205 cells, DC-SIGN affinity chromatography, and mass spectrometry were performed as described previously [18]. Briefly, the affinity column for soluble recombinant DC-SIGN-Fc (R&D Systems, Minneapolis, MN, USA) was prepared using Protein G sepharose and disuccinimidyl suberate cross-linker. COLO205 cells were suspended in hypotonic buffer [10 mM Tris-HCl (pH 7.6) and 0.5 mM $MgCl_2$ containing protease inhibitors], and homogenized using a Dounce homogenizer. The solution was restored to isotonic conditions through addition of NaCl solution. After centrifugation at $150,000 \times g$ for 45 min at 4 °C, the pellet was solubilized with lysis buffer [150 mM NaCl, 20 mM Tris-HCl (pH 7.5), 1 mM EDTA, and 1% Triton X-100 containing protease inhibitors], and centrifuged at $10,000 \times g$ for 60 min at 4 °C. The supernatant was retained as the membrane protein fraction, which was applied to the DC-SIGN-Fc column in the presence of $Ca^{2+}$. DC-SIGN ligand proteins were eluted with Tris-buffered saline (TBS) containing 10 mM ethylenediaminetetraacetic acid (EDTA). The eluate was re-applied to the column and a second elution step was conducted with TBS containing 50 mM mannose. The proteins eluted with mannose were subjected to sodium dodecyl sulfate–polyacrylamide gel electrophoresis (SDS–PAGE) on a 10% gel under reducing conditions. The gel was then stained with a silver staining kit (Wako). The DC-SIGN ligand bands were excised and digested with trypsin, and the fragments were analyzed via liquid chromatography–mass spectrometry (LC–MS/MS) with a hybrid quadrupole/time-of-flight spectrometer (Qstar pulsar I, Applied Biosystems, Foster City, CA, USA) interfaced to a Paradigm MS4 HPLC (Michrom BioResources, Auburn, CA, USA).

### 2.3. Co-Precipitation Assay, MSPL Protease Digestion Assay, and DC-SIGN Lectin Blot

HEK293 cells stably expressing full-length MSPL or TMPRSS13 were lysed with lysis buffer [150 mM NaCl, 50 mM HEPES (pH 7.4), and 5 mM $CaCl_2$ containing 1% NP-40]. The whole-cell lysate was incubated with DC-SIGN-Fc or IgG-Fc recombinant proteins, which were precipitated using Protein G sepharose beads (Thermo Fisher Scientific, Waltham, MA, USA) in the presence of $Ca^{2+}$. After washing with wash buffer (TBS containing 5 mM $CaCl_2$ and 0.05% Tween 20), the beads were suspended in elution buffer (TBS containing 10 mM EDTA) to chelate $Ca^{2+}$. The eluted solution was subjected to SDS–PAGE and western blotting using anti-FLAG M2 antibody. For the MSPL protease digestion assay, purified DC-SIGN–ECD (3 μg) and MSPL–ECD (0–1.0 μg) were mixed and co-incubated in the presence of $Ca^{2+}$ or EDTA for 0–6 h at 37 °C. To confirm the effect of N-glycosylation, MSPL–ECD was pre-treated with recombinant N-glycosidase F (PNGase F) (Roche, Basel, Switzerland). For this treatment, MSPL–ECD (1 μg) was dissolved in reaction buffer (TBS containing 0.5% SDS, 40 mM EDTA, and 1% 2-mercaptoethanol) and heated to 105 °C for 5 min. Then, PNGase F (3.2 units) was added to the MSPL–ECD solution, which was incubated for 24 h at 37 °C. The solution was treated with five times SDS sample buffer (2% SDS, 10% glycerol, 0.001% bromophenol blue, and 65 mM Tris-HCl, pH 6.8) and heated to 98 °C for 3 min under reducing conditions. The samples were separated via SDS–PAGE using a 5%–20% gradient gel, followed by Coomassie Brilliant Blue (CBB) staining. DC-SIGN lectin blotting was performed using DC-SIGN–ECD as the primary reaction solution. Reacted DC-SIGN–ECD on a

nitrocellulose membrane was incubated with anti-DC-SIGN monoclonal antibody (R&D Systems), followed by detection with HRP-conjugated anti-mouse IgG antibody.

### 2.4. Immunohistochemical Staining of COLO205 Cells and Human Colorectal Cancer Tissues

COLO205 cells were cultured on chambered cell culture slides (Corning, Inc., Corning, NY, USA) and fixed with phosphate-buffered paraformaldehyde (4%). For DC-SIGN staining, the cells were blocked with blocking buffer (TBS containing 10 mM $CaCl_2$ and 1% bovine serum albumin) and then incubated with DC-SIGN–ECD (0.7 μg/μL) in TBS containing 10 mM $CaCl_2$, 10 mM EDTA, or 50 mM mannose. The samples were incubated with anti-DC-SIGN monoclonal antibody (R&D Systems) followed by Alexa Fluor 546 secondary antibody. For anti-MSPL/TMPRSS13 antibody staining, rabbit polyclonal antibody against the TMPRSS13 catalytic domain (Abcam, Cambridge, UK), which reacts with both TMPRSS13 and MSPL, was used as a primary antibody solution, followed by visualization with Alexa Fluor 488 secondary antibody. A colorectal carcinoma-tissue array slide was obtained from SuperBioChips Laboratories (Seoul, Korea). After deparaffinization with xylene, the slide was hydrated with ethanol and immersed in 0.01 M citrate buffer. Then, the slide was boiled for 5 min in a microwave oven to obtain antigens. DC-SIGN staining and anti-MSPL/TMPRSS13 antibody staining were conducted as described above. All stained samples were observed using a Fluoview FV1000 confocal laser-scanning microscope (Olympus, Tokyo, Japan).

### 2.5. Flow Cytometry

U937 and U937-DC-SIGN cells were suspended in FACS buffer (PBS containing 2% FCS). To analyze DC-SIGN expression, the cells were incubated with primary anti-DC-SIGN monoclonal antibody (R&D Systems) diluted with FACS buffer. After washed with PBS three times, the cells were incubated with Alexa Fluor 488 anti-mouse IgG antibody. Then the cells were analyzed with BD FACSCalibur (BD Biosciences, CA, USA).

## 3. Results

### 3.1. Identification of MSPL/TMPRSS13 as DC-SIGN Ligands

We previously identified Mac-2BP as a DC-SIGN ligand expressed in COLO205 colon cancer cells based on recombinant DC-SIGN-Fc, a fused protein of the DC-SIGN extracellular domain and human IgG-Fc [18]. In the results of DC-SIGN-Fc affinity chromatography of the COLO205 membrane protein fraction, we found a sharp band at 100 kDa located above the Mac-2BP band (90 kDa) (Figure 1a). To identify this new DC-SIGN ligand protein, we analyzed the 100-kDa and 90-kDa bands using LC-MS/MS. The results showed that the band at 100 kDa and the broad band at 90 kDa contained the peptide sequence NKPGVYTK, corresponding to TMPRSS13 isoform 1 (MSPL) while Mac-2BP was detected at 90 kDa (Figure 1b) (see Tables S1 and S2). MSPL and TMPRSS13 are splicing variants of a single type II membrane serine protease, which was cloned from a human lung cDNA library [19]. When FLAG-tagged full-length MSPL and TMPRSS13 were expressed in HEK293 cells, each showed two bands (Figure 1d), presumably due to different post-translational modifications. Next, to validate the molecular interaction between DC-SIGN and MSPL/TMPRSS13, we conducted a DC-SIGN-Fc pull-down assay using cell lysates of MSPL and TMPRSS13 transfectants (Figure 1c–e). Western blot results indicated that both MSPL and TMPRSS13 were co-precipitated with DC-SIGN-Fc in the presence of $Ca^{2+}$ (Figure 1e). On the other hand, neither MSPL nor TMPRSS13 were co-precipitated by Fc only control, indicating the interaction was mediated by DC-SIGN and not by Fc domain. In addition, immunofluorescent staining for MSPL/TMPRSS13 showed intense cell-surface staining in HEK293 MSPL transfected cells (Figure 1f) and COLO205 cells (Figure 1g). Moreover, in COLO205 cells, MSPL/TMPRSS13 staining was overlapped significantly with DC-SIGN-Fc staining (Figure 1g). The chelation of $Ca^{2+}$ with 10 mM EDTA or the addition of mannose (50 mM) clearly abolished

DC-SIGN binding, suggesting that the interaction was mediated by the CRD of DC-SIGN. These results revealed that DC-SIGN binds to MSPL/TM PRSS13 expressed in COLO205 cells.

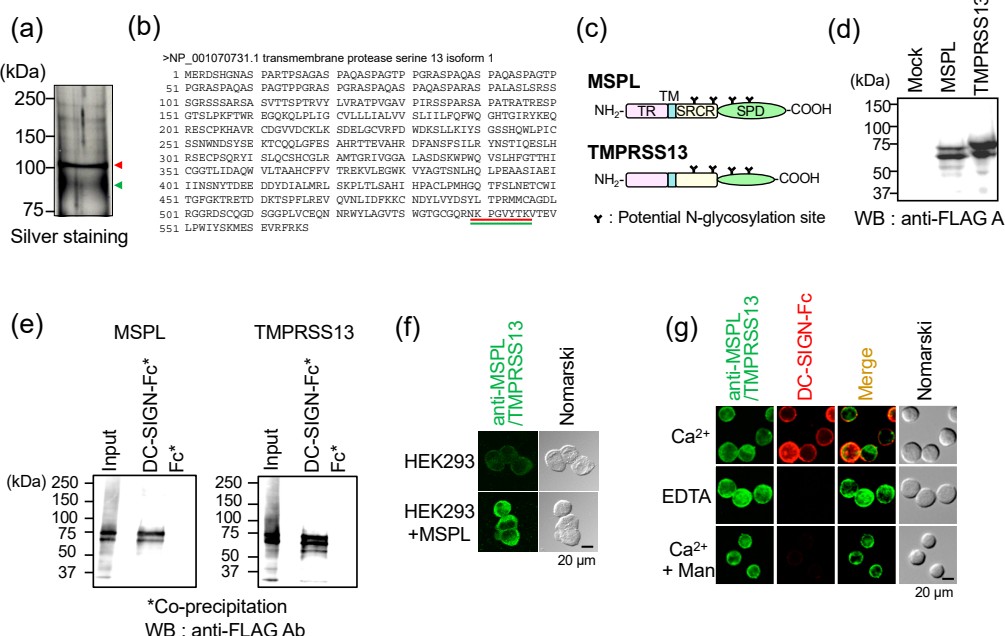

**Figure 1.** Identification of mosaic serine protease large-form (MSPL)/transmembrane protease serine 13 (TMPRSS13) as novel ligands of dendritic cell-specific intercellular adhesion molecule-3-grabbing nonintegrin (DC-SIGN) in colon cancer COLO205 cells. (**a**) DC-SIGN ligands were purified through affinity chromatography. The 90-kDa and 100-kDa protein bands were both excised and analyzed by liquid chromatography–mass spectrometry (LC-MS/MS). The arrowheads indicate proteins eluted from the affinity column. (**b**) Identification of TMPRSS13 through LC-MS. The detected peptide sequences are underlined with red (upper band) and green (lower band) lines. (**c**) Domain structures and potential N-glycosylation sites of MSPL and TMPRSS13. TR, tandem repeat domain; TM, transmembrane domain; SRCR, scavenger receptor cysteine-rich domain; SPD, serine protease domain. (**d**,**e**) Glycan-dependent interaction of DC-SIGN with recombinant MSPL and TMPRSS13 expressed in HEK293 cells. (**d**) HEK293 cells were transfected with full-length MSPL or TMPRSS13. The cell lysates were subjected to western blotting (WB) using anti-FLAG M2 antibody. (**e**) Lysates of MSPL and TMPRSS13 transfectant cells were incubated with DC-SIGN-Fc beads. Pulled-down proteins were subjected to Western blotting (WB) using anti-FLAG M2 antibody. (**f**) Staining of HEK293 MSPL transfectant cells with anti-TMPRSS13 antibody. (**g**) Double staining of COLO205 cells with DC-SIGN-Fc and anti-TMPRSS13 antibody.

### 3.2. Localization of MSPL/TMPRSS13 in Colorectal Carcinoma Tissue

The expression of MSPL/TMPRSS13 has been reported in normal tissues such as lung, skin, and prostate, but not in the colon [19,23]. To assess its expression in human colorectal cancerous and adjacent noncancerous tissues, we performed fluorescent immunohistochemistry (Figure 2). We found that MSPL/TMPRSS13 is highly expressed in various colorectal cancerous tissues but has lower expression in noncancerous tissues. Notably, we observed intense staining of MSPL/TMPRSS13 at the apical epithelial surface of various colorectal cancer tissues, whereas weak, broad staining was observed throughout the noncancerous mucosa. These results suggest that, during oncogenesis, localization of MSPL/TMPRSS13 shifts to the luminal side of the colon epithelium, where MSPL/TMPRSS13 obtains its DC-SIGN-bindable glycan structure.

The expression of MSPL/TMPRSS13 (green) and DC-SIGN ligands (red) was visualized through laser confocal microscopy. Yellow fluorescence indicates merged green and red signals, as shown in the right panels. Nomarski images are shown on the right side of each panel. Scale bars, 100 μm.

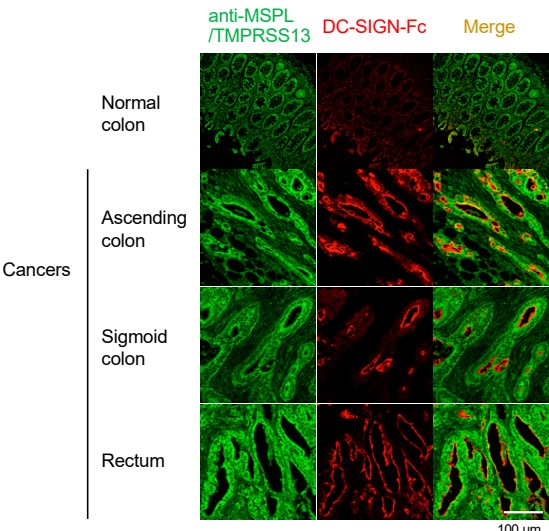

**Figure 2.** Expression of MSPL/TMPRSS13 in human colorectal carcinoma tissues.

### 3.3. Glycan-Dependent Recognition of MSPL by DC-SIGN

It has been reported that MSPL/TMPRSS13 retains its protease activity when it is phosphorylated, shed, and released from the cell surface [24]. Next, to explore the functional aspects of the molecular interaction between DC-SIGN and MSPL/TMPRSS13, we constructed a plasmid containing the MSPL extracellular domain (MSPL–ECD). A stable transfectant of MSPL–ECD showed multiple bands after CBB staining (Figure 3a), as previously reported [24], due to self-cleavage and glycosylation variants. When we conducted the DC-SIGN–ECD lectin blot assay, we found that DC-SIGN–ECD binds to MSPL–ECD in a $Ca^{2+}$-dependent manner (Figure 3b) and that all self-digested peptide fragments contained DC-SIGN-bindable glycans. Next, to test for direct involvement of N-glycans on MSPL–ECD in this molecular association, MSPL–ECD was treated with PNGase F. In CBB staining, most bands were shifted to lower molecular weights (Figure 3c, left), indicating the presence of several N-glycans in MSPL–ECD. Moreover, the DC-SIGN lectin blot showed that DC-SIGN binding was markedly attenuated with PNGase F treatment (Figure 3c, right). Together, these results demonstrate that MSPL recognition by DC-SIGN occurs in an MSPL N-glycan-dependent manner.

### 3.4. Glycan-Independent Digestion of DC-SIGN by MSPL

Next, to evaluate the effect of the MSPL–DC-SIGN interaction on MSPL protease activity, we performed a co-incubation study. When MSPL–ECD and DC-SIGN–ECD were co-incubated in the presence of $Ca^{2+}$, the band at 45 kDa in the DC-SIGN–ECD sample disappeared and three new bands were detected (Figure 4a, lane 3). This result indicates that DC-SIGN–ECD was degraded upon incubation with MSPL–ECD. To determine the optimal conditions for DC-SIGN digestion, we tested various MSPL–ECD concentrations (Figure 4b) and incubation times (Figure 4c). The results showed that DC-SIGN digestion was dependent on MSPL–ECD dose and time and incubation with 1 μg MSPL for 6 h was the optimal condition. We next performed a co-incubation study in the presence of EDTA (Figure 4d,e). The digested fragment of DC-SIGN–ECD showed a different pattern from that observed in the presence of $Ca^{2+}$, and one DC-SIGN–ECD band at 33 kDa overlapped with the MSPL band (Figure 4d, green and purple arrowheads). Notably, the full-length DC-SIGN–ECD band at 45 kDa disappeared with even a low level (0.5 μg) of MSPL–ECD, indicating that DC-SIGN–ECD degraded more efficiently when treated with EDTA. To confirm the DC-SIGN–ECD cleavage sites, we performed N-terminal amino acid sequencing analysis (Figure 4f). We identified the AAVGE sequence, which frequently appears in the DC-SIGN–Fc repeat domain, the SNRFTW sequence of the CRD, and the DYKDD sequence of the N-terminal FLAG-tag. These results clearly showed different cutting patterns of DC-SIGN–ECD in the presence or absence of $Ca^{2+}$. Given that DC-SIGN

binds to MSPL in a $Ca^{2+}$-dependent manner, the results of these co-incubation studies suggest that glycan-mediated MSPL recognition by DC-SIGN interferes with DC-SIGN digestion by MSPL (see Figure S1).

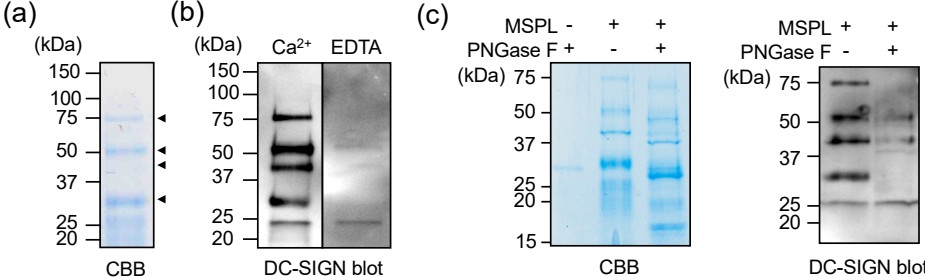

**Figure 3.** $Ca^{2+}$-dependent recognition of MSPL by DC-SIGN. (**a**) The extracellular domain of human MSPL (MSPL–ECD(extracellular domain)) with a C-terminal FLAG-tag was expressed in human hepatoma HLF cells. The secreted MSPL was purified and separated through sodium dodecyl sulfate–polyacrylamide gel electrophoresis (SDS–PAGE) under reducing conditions and detected through Coomassie Brilliant Blue (CBB) staining. (**b**) The extracellular domain of DC-SIGN (DC-SIGN–ECD) was prepared and applied to DC-SIGN lectin blotting. MSPL–ECD blotted on a nitrocellulose membrane was reacted with DC-SIGN–ECD in the presence of $Ca^{2+}$ or EDTA. (**c**) The purified MSPL–ECD was pretreated with PNGase F for 24 h at 37 °C and then subjected to DC-SIGN lectin blotting.

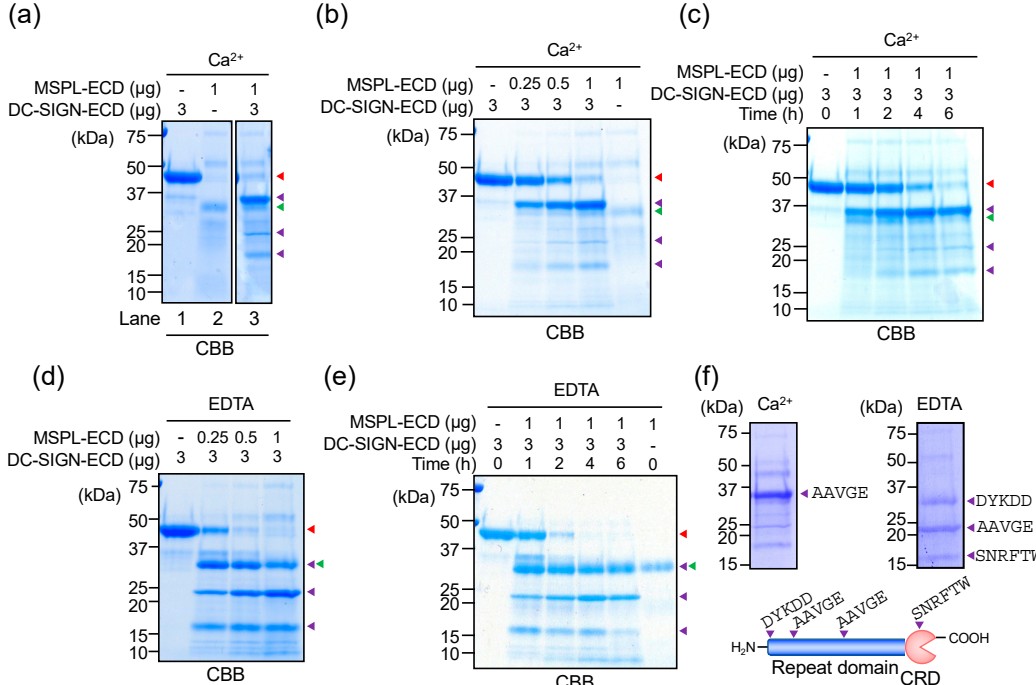

**Figure 4.** Proteolytic activity of MPSL in the presence of $Ca^{2+}$ and EDTA. (**a**) The purified MSPL–ECD and DC-SIGN–ECD were separated via SDS–PAGE under reducing conditions and detected with CBB staining (lanes 1 and 2). The two molecules were co-incubated in the presence of $Ca^{2+}$ at 37 °C for 6 h (lane 3). The colored arrowheads indicate bands of full-length (red) and digested DC-SIGN–ECD (purple), and MSPL–ECD (green). (**b–e**) The MSPL–ECD dose- (**b,d**) and time-dependency (**c,e**) of DC-SIGN degradation in the presence of $Ca^{2+}$ (**b,c**) and EDTA (**d,e**) were tested. (**f**) DC-SIGN–ECD cleavage sites were identified through N-terminal amino acid sequencing (BIOSUMS, Shiga, Japan). The amino acid sequences obtained are shown using one-letter codes. CRD: Carbohydrate recognition domain.

### 3.5. Cellular DC-SIGN as a Target of MSPL

The observation of glycan-dependent and -independent dual recognition systems between DC-SIGN and MSPL prompted us to test whether cellular DC-SIGN can also act as a substrate for MSPL protease activity. Therefore, full-length DC-SIGN was stably transfected into human U937 monocytic cells. Flow cytometry demonstrated that DC-SIGN was successfully transfected into the U937 cells (Figure 5a). We then extracted the membrane protein fraction of U937-DC-SIGN cells ($2.5 \times 10^6$ cells) and incubated it with MSPL–ECD protein (20 µg) at 37 °C for 6 h. Western blotting revealed that the DC-SIGN band shifted into two lower bands, indicating that full-length DC-SIGN was digested by MSPL–ECD (Figure 5b). These results demonstrated that full-length DC-SIGN is a substrate for MSPL.

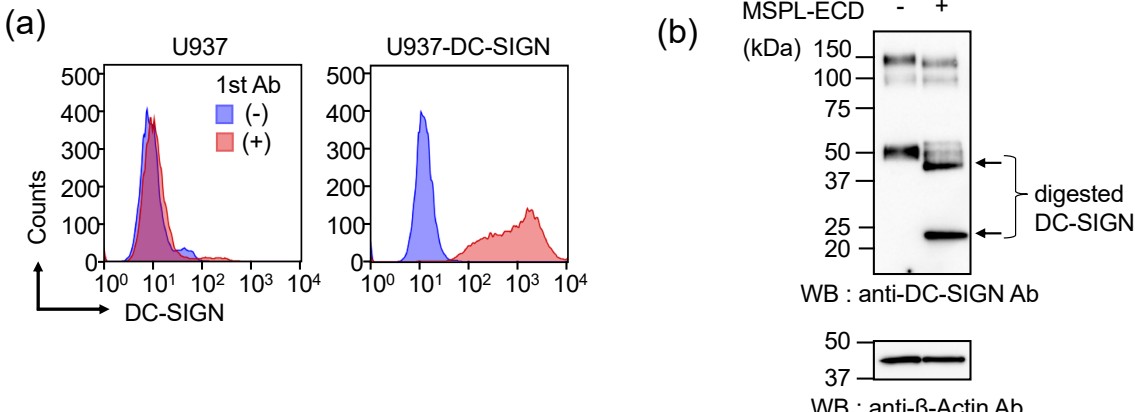

**Figure 5.** Degradation of DC-SIGN expressed in the monocytic cell membrane. (**a**) Expression of full-length DC-SIGN in the human monocytic U937 cell line. U937 and U937-DC-SIGN cells were incubated with primary anti-DC-SIGN monoclonal antibody (R&D Systems) followed by Alexa Fluor 488 secondary antibody, and analyzed through flow cytometry (BD FACSCalibur). (**b**) The membrane protein fraction was prepared from U937-DC-SIGN cells ($2.5 \times 10^6$ cells). The fraction was incubated with and without MSPL–ECD (20 µg) at 37 °C for 6 h. Western blotting was performed using anti-DC-SIGN monoclonal antibody or anti-β-actin antibody. The arrows in lane 2 indicate digested DC-SIGN fragments.

## 4. Discussion

Type II transmembrane serine proteases (TTSPs) are a large family, containing 17 proteases categorized into four subfamilies in humans. All subfamilies have a serine protease domain at the C-terminus where histidine, aspartate, and serine residues form a catalytic triad [20,25,26]. The substrates of TTSPs are cytokines, growth factors, and extracellular matrix components. Soluble forms of TTSPs are often detected in culture media, suggesting that their extracellular domains are shed from the cell surface [27–30]. Since this shedding is dependent on their protease catalytic activity [30], it is assumed to occur as a result of self-cleavage. Among TTSPs, the hepsin/transmembrane protease serine (TMPRSS) subfamily characteristically contains a group A scavenger receptor domain in the stem region. Mosaic transmembrane serine protease (MSPL) and TMPRSS13, which belong to the TMPRSS subfamily, are splicing variants of a single gene cloned from a human lung cDNA library [19,25]. MSPL/TMPRSS13 is highly expressed in the skin, lung, and bladder, but is not detected in colon tissue [19].

The physiological functions of MSPL and TMPRSS13 have been defined in epidermal barrier development [23] and virus infections [31–33]. By contrast, a number of reports describe other TTSP proteins as extensively associated with tumor growth and metastasis [26,34]. For example, TMPRSS1 is upregulated in several types of cancer, including prostate [35] and ovarian cancers [36], and is involved in cancer cell migration and invasion [37]. TMPRSS1 expression is linked to poor prognosis in prostate cancer patients, suggesting that TMPRSS1 serves as a biomarker for prostate cancer [38].

Here, we provide the first report that MSPL/TMPRSS13, which is in the same protein subfamily as TMPRSS1, is overexpressed in colorectal cancer cells. We demonstrated that MSPL/TMPRSS13 is recognized by DC-SIGN based on N-glycans and that this molecular interaction partially blocks MSPL/TMPRSS13 protease activity. Moreover, a previous report indicated that substrates containing Arg at the P1 position and Arg or Lys at position P2 are preferably cleaved by MSPL [25]. In contrast, the N-terminal amino acid sequence analysis in this study showed Leu-Lys and Ser-Arg at positions P1-P2 of the AAVGE and SNRFW sequences, respectively. These results indicate a novel substrate specificity of MSPL. Moreover, the physiological significance of the molecular interaction between DC-SIGN and MSPL/TMPRSS13 should be considered in light of its bidirectionality. In previous studies, we revealed that recognition of CEACAM-1 and Mac-2BP by DC-SIGN attenuates DC maturation, resulting in promotion of immune escape by cancer cells [17,18]. In this context, further investigation could reveal whether MSPL/TMPRSS13 recognition by DC-SIGN affects DC maturation. Meanwhile, MSPL/TMPRSS13 may be involved in cancer cell invasion and metastasis, as observed in other members of its subfamily, by degrading the basement membrane through protease activity. The physiological relevance of the direct involvement of MSPL/TMPRSS13 in tumor metastasis, including the influence of its interaction with DC-SIGN, requires further study.

Lewis (Le) glycans are the determinants of blood group antigens in glycolipids and glycoproteins of normal tissues such as erythrocytes and epithelial cells [39]. There are two types of Le glycans: type I Le glycans include Le$^a$ [Galβ1-3(Fucα1-4)GlcNAc] and Le$^b$ [Fucα1-2Galβ1-3(Fucα1-4)GlcNAc] glycans, while type II Le glycans are comprised of Le$^x$ [Galβ1-4(Fucα1-3)GlcNAc] and Le$^y$ [Fucα1-2Galβ1-4(Fucα1-3)GlcNAc] glycans. Aside from their basal expression in normal colon epithelium cells, Le glycans reportedly emerge during the course of oncogenesis, and are then referred to as tumor-associated Le glycans. We previously identified a unique Le glycan complex expressed in the colon cancer cell line SW1116 through affinity chromatography of mannan-binding protein (MBP), which includes a C-type lectin with specificity to type I, but not type II, Le glycans [40,41]. The glycan structure was found to be tetraantennary, containing N-glycans with β1-6 branching that harbor an unusual Le$^a$ tandem repeat and end with Le$^b$ at the nonreducing terminus. Our findings strongly suggest the presence of tumor-associated Le glycans in human colon tumors. Indeed, we observed strong expression of MBP high-affinity ligands in 38.5% of human colorectal carcinoma tissues, but not in adjacent nonmalignant tissues where blood-type Le glycans were expressed [42]. Thus, it is possible that MSPL expressed in colon cancer cells harbors complicated tumor-associated Le glycans. Because antibodies generally recognize a few oligosaccharides at most, lectins such as MBP and DC-SIGN that form large multimers can be developed into new diagnostic systems for detecting complex glycan structures in colorectal cancer.

## 5. Conclusions

Our study revealed the dual recognition system between DC-SIGN and MSPL/TMPRSS13 in colorectal cancer, providing novel insights into the mechanisms active in the tumor microenvironment. A comprehensive understanding of this system would help to achieve more effective diagnostics and treatment of colorectal cancer.

**Supplementary Materials:** The following are available online at http://www.mdpi.com/2076-3417/10/8/2687/s1, Figure S1: Dual recognition of DC-SIGN and MSPL, Table S1: Mascot search of the DC-SIGN ligand proteins, Table S2: Determination of MSPL in Mascot data.

**Author Contributions:** Conceptualization, T.K.; Methodology, N.K. (Nana Kawasaki); Software, N.K. (Nana Kawasaki); Formal analysis, N.K. (Nana Kawasaki); Investigation, S.M., M.N., and N.K. (Nobuko Kawasaki), resources, H.K.; Data curation, M.N. and T.K.; Writing—Original draft preparation, M.N. and S.M.; Writing—Review and editing, T.K. and N.K. (Nobuko Kawasaki); Visualization, S.M. and M.N.; Supervision, T.K.; Project administration, T.K.; Funding acquisition, T.K., B.Y.M., and M.N. All authors have read and agreed to the published version of the manuscript.

**Funding:** This research was supported in parts by a Grant-in-Aid for Scientific Research on Priority Areas and Creative Research A-14082203 (to T.K.), for Scientific Research, C-18590471 (to B.Y.M.) and for JSPS Fellows

10J09530 (to M.N.) from the Japan Society for the Promotion of Science, Ministry of Education, Culture, Sports, Science, and Technology of Japan, and by the R-GIRO (Ritsumeikan Global Innovation Research Organization) Program (to T.K.).

**Conflicts of Interest:** The authors declare no conflict of interest.

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
