# Peer review of "Glycan-Dependent and -Independent Dual Recognition between DC-SIGN and Type II Serine Protease MSPL/TMPRSS13 in Colorectal Cancer Cells"

_applsci, doi:10.3390/app10082687_

Round 1
Reviewer 1 Report
The manuscript entitled “Glycan-dependent and -independent dual recognition between DC-SIGN and Type II serine protease MSPL/TMPRSS13 in colorectal cancer cells” by Motohiro Nonaka and co-workers reported here a dual recognition phenomenon between a C-type lectin DC-SIGN and a type II transmembrane serine protease MSPL as long as its splice variant TMPRSS13. The authors identified MSPL/TMPPRSS13 as a ligand for DC-SIGN through affinity chromatography coupled with DC-SIGN-Fc. The authors further verified this binding using Co-IP and immunohistochemical staining. The authors found the binding between DC-SIGN and MSPL is calcium dependent, and further confirmed it is N-glycan-dependent. However, in another assay, which shown MSPL can digest DC-SIGN in a glycan-independent manner. Thus, the authors concluded that DC-SIGN and MSPL interact with each other in a glycan-dependent and independent manner. The authors provided experiments evidence to support this conclusion. I still have some major and minor concerns below:
Major concerns:
1). The authors need to do an important control for their immunohistochemical staining experiments showing in Figure 1f, which is to include a negative control (cells do not overexpress MSPL/TMPRSS13). The current results can not support DC-SIGN direct binds to MSPL on cell surface.
2). To show DC-SIGN binds to MSPL in a calcium dependent manner, the authors need to do another control experiment, which is to reduce the calcium to 0.1 mM Ca2+ (no EDTA) instead of no calcium to repeat the result in Figure 3b.
3). Did the author do the experiments in Figure 3b and Figure 4 using similar buffer recipe? Is it possible the binding of MSPL and DC-SIGN not only depend on calcium, but also some other conditions like salt concentration, pH?
4). The authors showed MSPL can cleave DC-SIGN independently of calcium. How do the authors exclude the possibility that the DC-SIGN-Fc antibody used to detect DC-SIGN in the immunohistochemical staining experiments simply can not detect cleaved product of DC-SIGN in calcium free condition?
Minor concerns:
1). Materials and Methods need to add paragraphs for description the methods used to do the incubation experiments of MSPL and DC-SIGN and also the flow cytometry experiments.
2). On page 1, line 29, “serine protease MSPL and its splice variant TMPRSS13 as DC-SIGN receptors….” should be written as “serine protease MSPL and its splice variant TMPRSS13 as DC-SIGN ligands….”.
3). On page 4, line 148, “then incubated with DC-SIGN-ECD (0.7 μg/μl) in TBS containing 10 M CaCl2…. “ . It is most likely 10 mM CaCl2, correct it.
4). On page 5, Figure 1 legends is not corresponded to the actually figure. Please correct this error.
5). The authors need to provide the LC-MS sequence coverage file for the identification of MSPL/TMPRSS13. It is better to see the mass spec detect multiple peptides along MSPL, besides “NKPGVYTK”.
6). What could be the explanations that the bands contain MSPL/TMPRSS13 from the affinity chromatography and overexpressed HEK293 cells show obvious molecular weight differences, with the ones from HEK293 experiments are smaller?
Reviewer 2 Report
The manuscript by Nonaka et al describes identification and characterisation of physical interactions between the MSPL/TMPRSS13 protease and DC-SIGN. The manuscript presents a large amount of high quality data, and is generally convincing, especially regarding characterisation of the protease activity of MSPL towards DC-SIGN in vitro. However, I have some concerns that could benefit from clarification, particularly regarding the identification of MSPL/TMPRSS13 as bona fide in vivo interactors of DC-SIGN.
Major comments:
1) I have some concerns with the representation of the co-IP data shown in Figure 1.
L165 / Figure 1A. Was this band present in negative control pulldown experiments using only Fc?
Please show the data demonstrating identification of the NKPGVTYK peptide.
L175. Please describe the Fc only pulldown controls in more detail in the text.
Ideally, please show full co-IP Western blots, showing detection of MSPL or TMPRSS13 in the whole extract as well as the co-IPed fraction.
Has co-IP been tested in the presence of Ca2+ + mannose, or in the presence of EDTA? This would substantially strengthen the evidence for DC-SIGN dependent interaction.
Figure 1C. Describe TR, TM, SRCR, and SPD in the legend.
2) In the cell staining shown in Figure 1(f), are the cells expressing MSPL or TMPRSS13? Please correct/clarify, and ideally show both, individually. How are MSPL/TMPRSS13 visualised in this experiment? What does staining for "DC-SIGN ligands" exactly mean?
3) Figure 2. Please clarify what exactly is meant by "DC-SIGN ligands"? Is this DC-SIGN? How are MSPL/TMPRSS13 detected?
Minor comments:
4) Some errors in English expression throughout.
5) Abstract: "DC-SIGN recognizes these glycoproteins through N-glycans in a Ca2+ dependent manner." While DC-SIGN can recognise N-glycans on tumour cells and HIV, M. tuberculosis does not have N-glycans. Please correct/clarify this statement.
6) Paragraph from L53-62 needs extensive additional citations.
7) Abstract and L75
"Meanwhile, the soluble recombinant extracellular domain of DC-SIGN (DC-SIGN-ECD) was cleaved via MSPL/TMPRSS13 protease activity, indicating glycan-independent recognition of DC-SIGN by MSPL/TMPRSS13. "
It is not clear why this result implies the interaction is glycan-independent. Please re-phrase and clarify.
8) Please correct panel numbering in the legend of Figure 1.
9)L316 "Notably, DC-SIGN recognition is considered to be specific to MSPL/TMPRSS13, as no other TTSP family members were detected in the DC-SIGN affinity chromatography or MS analyses."
Remove this statement. Full details of the pulldown and MS experiments are not provided.
Round 2
Reviewer 1 Report
The authors did a great job to address all my concerns. I agree to publish the current version, but please fix figure 4.
I think the problem with Figure 4 is more likely due to formatting.
The authors need to make sure all the items on Figure 4 is properly displayed. Right now, the items are covered by some blank color backgrounds, it is hard to see anything on the figure. Hope this helps!
Author Response
Comments from Reviewer 1:
"I think the problem with Figure 4 is more likely due to formatting.
The authors need to make sure all the items on Figure 4 is properly displayed. Right now, the items are covered by some blank color backgrounds, it is hard to see anything on the figure. Hope this helps!"
Reply to Reviewer 1:
Thank you for your additional comments on Figure 4.
However, we could not reproduce the problems, which were pointed out by the reviewer, on our display but also on our printout.
According to the reviewer’s comment, we had checked Figure 4 very carefully and we found that everything is OK. So, we concluded that it is appropriate to keep everything as it was.
Incidentally, we found one mistype in Page 2, L44. We have changed "which require calcium for glycan recognition” to "which requires calcium for glycan recognition"(addition of ”s”).
We hope this revised manuscript will be accepted for publication in the special issue “Glycosylation in Cancer” of Applied Sciences as a regular paper.